

# IoT-IIRS: Internet of Things based intelligent-irrigation recommendation system using machine learning approach for efficient water usage

Ashutosh Bhoi[1], Rajendra Prasad Nayak[1], Sourav Kumar Bhoi[2], Srinivas Sethi[3], Sanjaya Kumar Panda[4], Kshira Sagar Sahoo[5] and Anand Nayyar[6]

[1] Department of Computer Science and Engineering, Government College of Engineering (Govt.), Kalahandi, India
[2] Department of Computer Science and Engineering, Parala Maharaja Engineering College (Govt.), Berhampur, India
[3] Department of Computer Science Engineering and Applications, Indira Gandhi Institute of Technology (Govt.), Sarang, India
[4] Department of Computer Science and Engineering, National Institute of Technology (NIT), Warangal, India
[5] Department of Computer Science and Engineering, SRM University, Amaravati, Andhra Pradesh, India
[6] Graduate School; Faculty of Information Technology, Duy Tan University, Da Nang, Viet Nam

## ABSTRACT

In the traditional irrigation process, a huge amount of water consumption is required which leads to water wastage. To reduce the wasting of water for this tedious task, an intelligent irrigation system is urgently needed. The era of machine learning (ML) and the Internet of Things (IoT) brings it is a great advantage of building an intelligent system that performs this task automatically with minimal human effort. In this study, an IoT enabled ML-trained recommendation system is proposed for efficient water usage with the nominal intervention of farmers. IoT devices are deployed in the crop field to precisely collect the ground and environmental details. The gathered data are forwarded and stored in a cloud-based server, which applies ML approaches to analyze data and suggest irrigation to the farmer. To make the system robust and adaptive, an inbuilt feedback mechanism is added to this recommendation system. The experimentation, reveals that the proposed system performs quite well on our own collected dataset and National Institute of Technology (NIT) Raipur crop dataset.

# INTRODUCTION

Water is the essential natural resource for agriculture, and it is limited in nature (*Kamienski et al., 2019*; *Wang et al., 2020*; *Sahoo et al., 2019*). In a country like India, a huge share of water is used for irrigation (*Nawandar & Satpute, 2019*). Crop irrigation is a noteworthy factor in determining plant yield, depending upon multiple climatic conditions such as air temperature, soil temperature, humidity, and soil moisture

Corresponding author
Anand Nayyar,
anandnayyar@duytan.edu.vn

(*Bavougian & Read, 2018*). Farmers primarily rely on personal monitoring and experience for harvesting fields (*Glaroudis, Iossifides & Chatzimisios, 2020*). Water needs to be maintained in the field (*Liu et al., 2020*; *Fremantle & Scott, 2017*). Scarcity of water in these modern days is a hot issue. Such a scarcity is already affecting people worldwide (*LaCanne & Lundgren, 2018*; *Schleicher et al., 2017*). The situation may worsen in the coming years.

The uniform water distribution of traditional irrigation systems is not optimal. Hence, research effort has been exerted towards efficient agricultural monitoring system (*Kim, Evans & Iversen, 2008*). In this regard, standalone monitoring station is under development. For instance, a mixed signal processor (MSP430) has been developed with a microcontroller along with a set of meteorological sensors. A wireless sensor-based monitoring system has also been developed. It has made of several wireless sensor nodes and a gateway (*Gutiérrez et al., 2013*). It has been implemented as an easy solution with better spatial and temporal determinations. In addition, there is a demand for automating the irrigation system. Automation enables machines to apply intelligence by reading historical data and accordingly analyze and predict the output. This mechanism is more effective than the traditional rule-based algorithm (*Chlingaryan, Sukkarieh & Whelan, 2018*). From here onwards, the role of machine learning (ML) and artificial intelligence (AI) play an important role. The applications of ML in the field of agriculture domain is numerous (*Jha et al., 2019*). Starting from crop selection and yielding to crop disease prediction, different ML techniques like artificial neural networks (ANN), support vector machine (SVM), k-nearest neighbor (k-NN), and decision trees have shown huge success (*Ge et al., 2019*). After the great success of the combination of WSN and ML techniques, there is a requirement for more automation and without human intervention. The development of machine to machine (M2M) and the Internet of Things (IoT) allows devices to communicate with one another without much human intervention. These days, the usage of mobile devices is increasing, at the same time cloud computing is becoming a popular technology. The existing water monitoring system used wireless sensors for monitoring the soil condition for irrigation. These systems barely capture the data from the land and subsequently controls the electric motor for watering the land.

Moreover, there is a high demand to analyze the real-time data based on the historical information for irrigating fields. Research on M2M system is limited, particularly regarding communication among devices to more intelligently perform analysis and recommendation. Realizing the water scarcity issue and at the same time the technological advancement, we are motivated to design a fully automated irrigation system. This system must be smart enough so that it can adapt to the local climatic conditions and precisely predict the decision on irrigation in a reliable way.

There are many aspects of an efficient automatic irrigation system. As weather data is an important parameter for making irrigation decisions, the system must be smart enough to integrate the forecasted weather data (*Goldstein et al., 2018*). The next important aspect is to estimate the soil features properly so that the prediction error can be minimized in the irrigation recommendation system (*Khoa et al., 2019*). Once the weather data along with soil and environmental parameters are forecasted, the data can be used to make final decisions regarding irrigation. For this task, we need an efficient binary classifier to

decide whether to irrigate or not. The advanced ML techniques and IoT may provide a solution. The two main motivations for this work are efficient water usage in irrigation to reduce water wastage and minimize human intervention as much as possible.

The main objective of the work is to reduce water wastage in the agricultural field by introducing an IoT-enabled ML-trained recommendation system (IoT-IIRS). In this prototype, the analyzed information can be sent from the cloud server to the farmer's mobile handset priorly. This system makes the decision of whether to water the field or not simple for farmers. The main contributions of this work are as follows:

1. An IoT-IIRS is proposed for efficient water usage.
2. IoT devices are deployed in the crop field to collect the ground and environmental details precisely. The gathered data such as air temperature, soil temperature, humidity, and soil moisture are forwarded through an Arduino and stored in a cloud-based server, which applies ML algorithms such as SVM, regression tree, and agglomerative clustering to analyze those data and suggest irrigation to the farmer.
3. To make the system robust and adaptive, an inbuilt feedback mechanism is added to this recommendation system.
4. Experimentation reveals that the proposed system performs well on our own collected dataset and NIT Raipur crop dataset.

The remaining part of the paper is organized as follows. The next section introduces the related work. The proposed methodology section discusses the working steps for the solution framework. The results of the experiments are discussed in the result section. Finally, the last section ends with concluding remarks and further scope for improvement.

## RELATED WORK

Communication is one of the most important aspects of the implementation of an automatic and intelligent irrigation system. In the last few years, significant research is carried out on smart irrigation. In *Salas et al. (2014)*, the authors suggested the implementation of general packet radio service (GPRS) communication as a gateway between wireless sensor network (WSN) and the Internet. Numerous data transmission techniques have been applied in closed-loop watering systems. These are used to apply the required amount of water in the desired place in due time to conserve natural resources. Bhanu et al. presented a system to build a wireless sensor network-based soil moisture controller, which determines the water demand by differentiating the soil moisture with a preset threshold value (*Bhanu, Hussain & Ande, 2014*). Field authentication tests are regularly performed on distinct soils to estimate the soil moisture and water quantity in the soil to develop a productive watering system. If the preserved data do not match with the measured soil data, then an interrupt is passed to the pressure unit to stop the irrigation. Solar power-based intelligent irrigation system is proposed to provide the required amount of water to the crop field (*Rehman et al., 2017*). Soil and humidity sensors are deployed to measure the wet and dry states of the soil. Once sensing is completed, the sensor node transmits the signal to the microcontroller. In addition, they forward that

signal to the relay to switch on and off the motor. In *Kaur & Deepali (2017)*, the authors presented a WSN-based smart irrigation system for efficient water usage with the help of automated remote sensing and persistent analysis of soil parameters and environmental conditions using ML. Hema et al. proposed an approach to estimate the native real-time weather parameters for interpolation with the help of an automated weather station (ASW) (*Hema & Kant, 2014*). This intelligent system provides past, present, and future predictions utilizing nearby ASW data and control the irrigation process during conditions like rainfall. To control the irrigation, soil moisture and ASW data are exploited for error correction, where interpolated value is compared with soil moisture data. In *Ashwini (2018)*, the authors stated that the smart irrigation system employing WSN and GPRS modules optimizes water utilization for any agricultural crop. This approach comprises of a distributed WSN along with different sensors, such as moisture and temperature sensors. Gateway components are employed to transfer data from the sensor unit to the base station. Direct order is sent to the actuator for regulating the irrigation process and handling data from the sensor unit. According to the need and conditions of the field, different algorithms are used in the system. It is programmed in a microcontroller that sends commands through an actuator to regulate water quantity with a valve unit. The entire framework is powered by photo-voltaic panels, where duplex communication takes place through cellular networks. Web applications control irrigation through regular monitoring and irrigation scheduling. In *Leh et al. (2019)*, the authors designed hardware and software by analyzing the routing protocols of the sensor network. Mobile phones and wireless personal digital assistant (PDA) help to monitor the soil moisture content, as a result, irrigation system is controlled. Mohapatra et al. proposed an irrigation system based on WSN (*Mohapatra, Lenka & Keswani, 2019*). The designed system uses fuzzy logic and neural network to save water efficiently. The used fuzzy neural network is an integrated set of fuzzy logic reasoning and self-learning ability of a neural network. Sensor nodes measure temperature, humidity, soil moisture, and light intensity data. LAN or WAN helps to transfer the collected data to the irrigation control system via gateway nodes. The electromagnetic valve is controlled for precision irrigation based on the collected data. To predict the soil moisture, the authors in (*Goap et al., 2018*), developed an algorithm that works based on field sensor and weather forecasting data. The algorithm uses the support vector regression model and k-means clustering. This algorithm also provides a suggestion regarding irrigation based on the level of soil moisture. Collected device information and the output of the algorithm are stored in MySQL Database at the server end. In *Gutiérrez et al. (2013)*, the authors presented a system that uses a camera to capture images. Captured images are processed to determine the water content of the soil. Depending on the level of water in the soil, water is pumped into the crop field. The camera is controlled from an Android application. The camera captures RGB pictures of soil using an anti-reflective glass window to find the wet and dry areas. The WiFi connection of the smartphone is used to transmit the estimated value to the gateway through a router to control the water pump. M2M communication is applied as a robust mechanism for effective water management during farm irrigation (*Shekhar et al., 2017*). In *Vij et al. (2020)*, the authors proposed a distributed network environment using IoT,

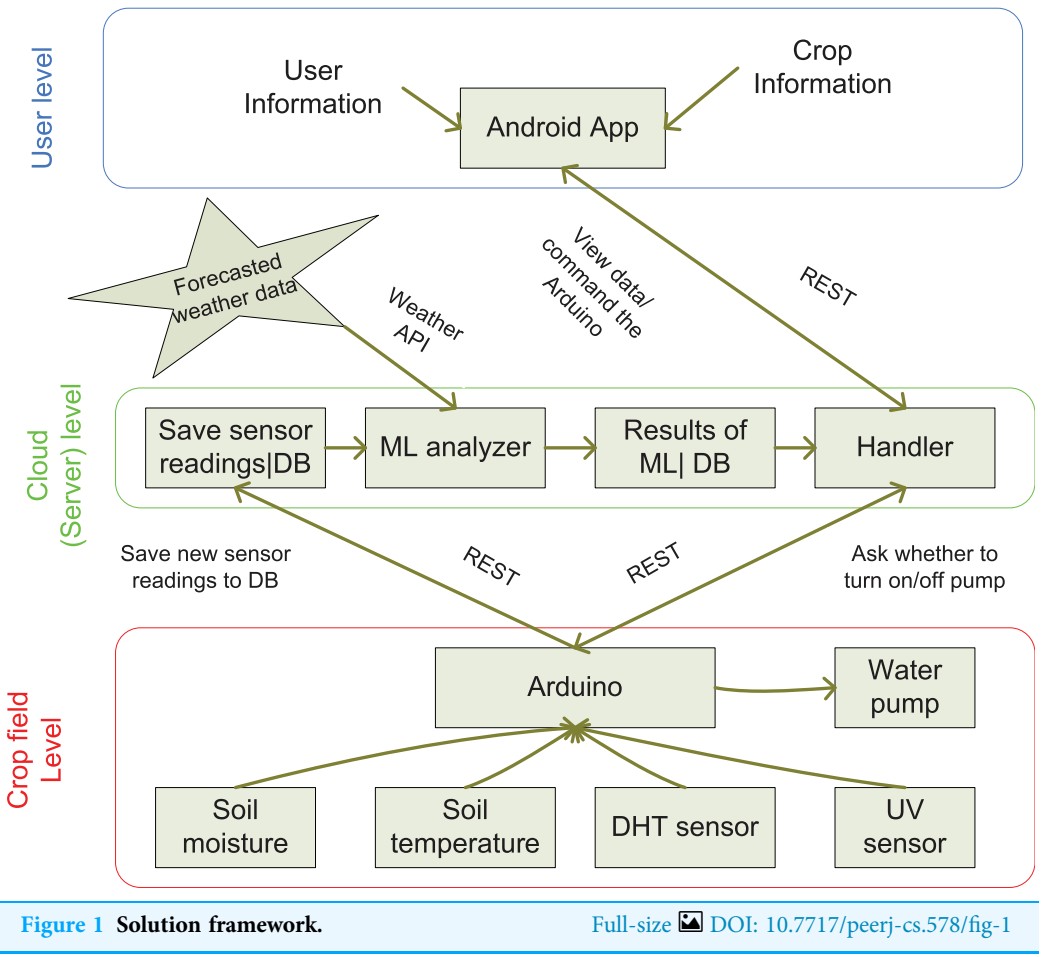

**Figure 1 Solution framework.**

ML, and WSN technologies for efficient water usage and reduced soil erosion. Soil moisture prediction is one of the most important tasks for an automatic irrigation system. Many researchers contributed various methodologies and algorithms for this task (*Adeyemi et al., 2018*; *Sinwar et al., 2020*; *Singh, Kaur & Kumar, 2020*).

## PROPOSED METHODOLOGY

The proposed three-tier architecture along with the descriptions are mentioned in this section. The solution architecture of our smart irrigation system is given in Fig. 1. The details of each level are described below.

• **Crop Field Level**: The first one is the crop field level where different sensors are deployed in the field. Various sensors like soil moisture (EC- 1258), soil temperature (DS18B20), air temperature (DHT11), and humidity (DHT11) are used to gather all these soil and environmental attributes. All these sensors collect data and send them to the Arduino. The Arduino then forwards the sensor data to the cloud server. All these sensor data are collected twice a day and forwarded for cloud storage using a micro-controller device. The average value is calculated and stored as the final reading for the particular day. To eliminate the inter-dependency among the used parameters, the Pearson correlation is computed. It is found that there is no strong correlation exists. The Arduino is also

connected to the motor pump through the breadboard and relay switch to turn on/off the pump for irrigation. Here, we use the Arduino microcontroller because it requires low energy.

- **Cloud Level**: The second level is the cloud level where the cloud server is used to provide service to the user. The sensors' data are stored in the database. The data are then feed to the ML-based model for analysis. This ML unit is the heart of this intelligent system, which has two sections. One is the regression model that is used to predict the soil and environmental parameters in advance. By doing so, it can be used effectively to improve the performance of the system. The parameters that are considered from forecasted weather data are the atmospheric pressure, precipitation, solar radiation, and wind speed. These predicted values are passed through a clustering model to reduce the predicted errors. The other ML-based model takes the results of the clustering model along with the forecasted weather data as input. This binary classification model categorizes the predicted samples into two predefined classes: irrigation required (Y) or not required (N). The results of these ML models are stored in the database for future actions. The last component of the cloud-based server is the handler used for coordination between the user and field units. Based on the suggestion of the ML model, the handler will send irrigation suggestions to the user via the Android application. Based on the agricultural literature, the formula used for calculating the water requirement is discussed in Eq. (1).

$$EV_o * C_f = W_{need} \qquad (1)$$

where, $EV_o$ = rate of evaporation
$C_f$ = crop factor
$W_{need}$ = amount of water needed.

- **User Level**: At the user level, the user interacts with an Android application to enter the details about the farmer and crop. Farmer credentials are used to authenticate the user through login operation. In crop details, the farmer may have to provide the information by selecting the drop-down menus like a session, crop name, total crop days, date of sowing, and so on. Through this application, the farmer may get all relevant information about the crop and field. Thus, upon receiving the sensor data along with the irrigation suggestion on the Android application, the user may order the on/off command to the microcontroller. Here, the system is an adaptive one that takes feedback from the user for each suggestion by the handler. If the farmer does not follow the recommendation, then feedback is sent to the server for updation. The system will be fine-tuned subsequently based on user feedback. The microcontroller upon receiving the on/off command from the user performs the motor on/off operation for the supply of water to plants. Thus, we have an automatic irrigation system, which can be used to increase the productivity of the crop by providing an optimal amount of water. The working steps of the proposed solution framework have shown in Algorithm 1.

**Algorithm 1 Working steps for solution framework.**

**Input:** Authentication details for Login to Android application; All relevant crop information from the drop down list

**Output:** Motor turn on/off

1: Collect all sensor data at regular intervals through the Arduino.

2: Save the sensor readings in the cloud server database.

3: Using the weather forecasted data, the ML analyzer model analyzes these stored sensor data to check whether irrigation is required or not?

4: The recommendation of the ML model is forwarded to the Android through the handler.

5: Based on the ML recommendation and sensor readings the user will inform the Arduino to send an on/off signal to the motor.

6: User may follow the recommendation and irrigate the field.

7: If the user does not follow the recommendation, then feedback will be sent and stored in the database for the corresponding sensor readings.

## ML model

As mentioned earlier, the ML analyzer is the main building block of our proposed system. On the stored sensor data, the regression tree (RT) algorithm is applied to predict future soil and environmental data. RTs may grasp non-linear relationships and are reasonably robust to outliers. These predicted parameters are further improved using the agglomerative clustering (AC) algorithm. It may be suitable for this task as the clusters are not supposed to be globular. The forecasted weather data are combined with the predicted soil data to be fed into the classification model. Then, the classifier categorizes the data sample whether irrigation is required or not at that time. This way, the ML technique helps the farmer with the suggestion for irrigation on the crop field. The details about this recommendation architecture is mentioned in Fig. 2. The working steps of the ML model are detailed in Algorithm 2.

### RT

RT is constructed via a procedure known as binary iterative splitting, which is a repetitive process that partition the data into branches, subbranches, and so on. Each decision node in the tree assesses the value of several input variable's values. The leaf nodes of the RT carry the predicted output (response) variable (*Torres-Barrán, Alonso & Dorronsoro, 2019*). Here, the AdaBoost algorithm is applied using its library.

### AC

AC is a type of hierarchical clustering that works on a bottom-up approach (*Stashevsky et al., 2019*). A fundamental assumption in hierarchical AC is that the merge operation is monotonic. Each scrutiny begins in its own cluster, and clusters merged as one moves up the hierarchy. This clustering may improve the performance of classical regression by partitioning the sample training space into subspaces.

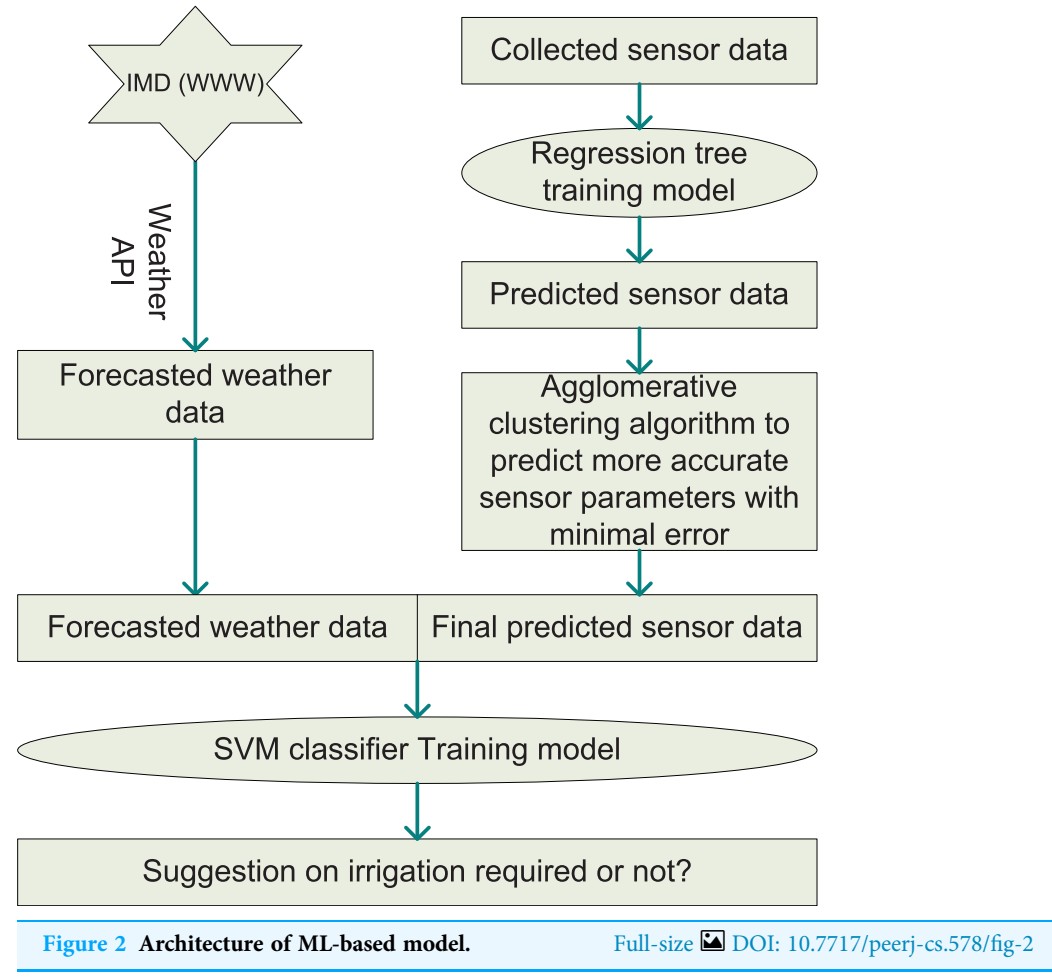

**Figure 2 Architecture of ML-based model.** 

### SVM

SVM is a supervised ML model that works very well for many classification tasks (*Lebrini et al., 2019*). Once SVM is fed with sets of labeled training data for each class, they can be categorized into new samples. For non-linear classification, it performs well with a limited number of labeled training data.

The prototype model and its deployment in the paddy field are depicted in Figs. 3A and 3B separately. It contains all sensors, Arduino, GSM module, different connecting wires. Along with all these IoT devices, one power bank is used as the source of power. This power source may be replaced with a small solar panel with a battery in the future. The Arduino is used because of its low power consumption. The prototype of the android application is illustrated in Fig. 4A and the recommendation for the motor using the classification, of whether irrigation is required or not is shown in Fig. 4B.

## RESULTS AND DISCUSSION

The performance of the proposed model has been evaluated using Python. All the ML models have been implemented using the Scikit-learn Python library. To validate the effectiveness of the system, our own collected data (GCEKIoTCommunity) (https://github.

**Algorithm 2** Working steps for ML model.

**Input:** All sensor data and the forecasted weather data from Indian Meteorological Department (IMD)

**Output:** Recommendation for irrigation required or not

1: Apply previously trained RT model on these sensor data to predict future soil and environmental data.

2: Apply AC algorithm for a more precise prediction of all these sensor parameters.

3: All predicted sensor parameters are combined with the forecasted weather data to prepare the final data samples.

4: Apply the previously trained SVM model on these final data samples.

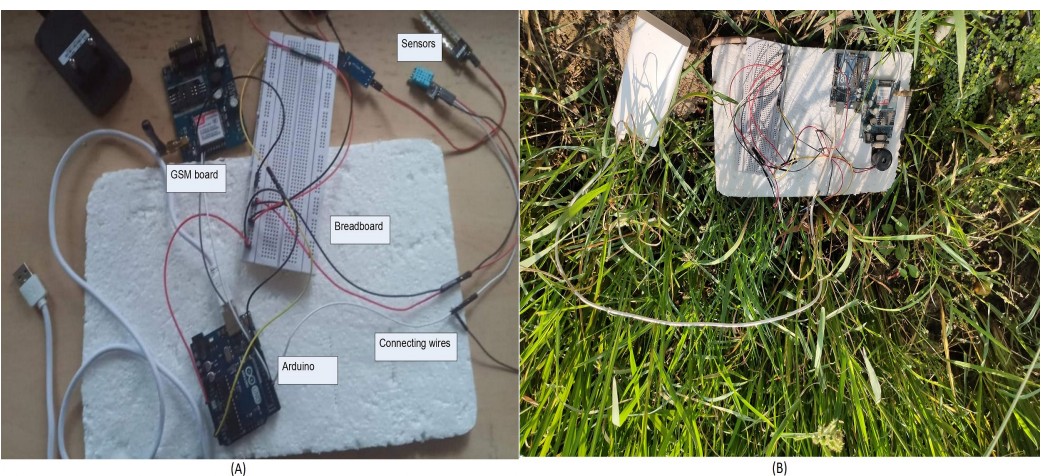

(A)  (B)

**Figure 3** (A) Prototype model (B) Model deployment inside paddy field.

com/NitrrMCACommunity/Agriculture-Automation-) along with the Crops dataset of NIT Raipur (NitrrMCACommunity) (https://github.com/NitrrMCACommunity/Agriculture-Automation-/tree/master/Dataset) have been used. We have collected data like soil moisture, soil temperature, humidity, air temperature, and UV radiation by using the relevant sensors. The stored readings of each day are the average of all readings on that day. We gathered 150 samples from our simulation area. In the NIT Raipur dataset, there are 501 samples available. As mentioned earlier, we have integrated the forecasted weather data from Indian meteorological department site (IMD) (https://mausam.imd.gov.in/) by using weather application programming interface (APIs) (https://restapitutorial.com/). Through this, we have collected the data like next-day rainfall, amount of rainfall, precipitation, and so on of a particular area to make our irrigation accuracy even better. As the evening is the best time to irrigate, each day in the evening, the ML model runs. Based on its results, the handler sends irrigation suggestions to the farmer along with the stored and forecasted parameters. For the classification task, the five-fold cross-validation method is used for better generalization. The efficiency of the system is estimated based on the most extensively used measures, such as precision, recall, F1-measure, and accuracy (A). Mathematically, these measures are represented in Eqs. (2) to (4).

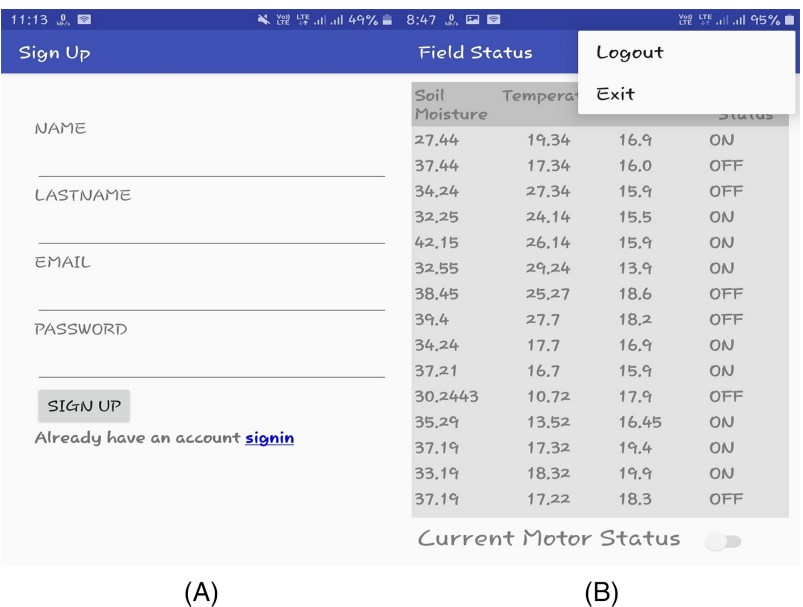

**Figure 4** (A) Sign-up page of the android application; (B) field status and logout page of the Android application.

**Table 1 Suggestion for irrigation.**

| Dataset | Classifier | 70:30 ratio of training and testing | | | | 5-fold cross validation | | | |
|---------|-----------|------|--------|--------|--------|------|--------|--------|--------|
| | | A | $P_r$ | $R_c$ | F1 | A | $P_r$ | $R_c$ | F1 |
| Our sensor collected data | Naïve Bayes | 83.48 | 82.67 | 80.95 | 81.80 | 83.61 | 82.77 | 80.99 | 81.87 |
| | Decision Tree (C4.5) | 85.74 | 85.12 | 83.81 | 84.46 | 85.83 | 85.19 | 83.88 | 84.53 |
| | SVM | 87.29 | 86.77 | 85.42 | 86.09 | 87.45 | 86.85 | 85.51 | 86.17 |
| NIT Raipur Crop dataset | Naïve Bayes | 84.37 | 83.35 | 82.63 | 82.99 | 84.51 | 83.44 | 82.68 | 83.06 |
| | Decision Tree (C4.5) | 86.15 | 85.75 | 83.89 | 84.81 | 86.29 | 85.86 | 83.95 | 84.89 |
| | SVM | 88.05 | 87.44 | 86.52 | 86.98 | 88.22 | 87.55 | 86.59 | 87.07 |

$$Precision\ (P_r) = \frac{T_p}{T_p + F_p} \tag{2}$$

$$Recall\ (R_c) = \frac{T_p}{T_p + F_n} \tag{3}$$

$$F1 - measure\ (F1) = \frac{2 * P_r * R_c}{P_r + R_c} \tag{4}$$

where $T_p$, $F_p$, and $F_n$ are denoted as true positives, false positives, and false negatives, respectively. The results of different ML models for classification are given in Table 1. The different performance evaluations of the proposed model on our own collected dataset are shown in Fig. 5A and 5B respectively. Similarly, the performance evaluation of the proposed model on NIT Raipur crop dataset is depicted in Figs. 6A and 6B respectively. The experimental results show that the performance of our system is quite satisfactory for

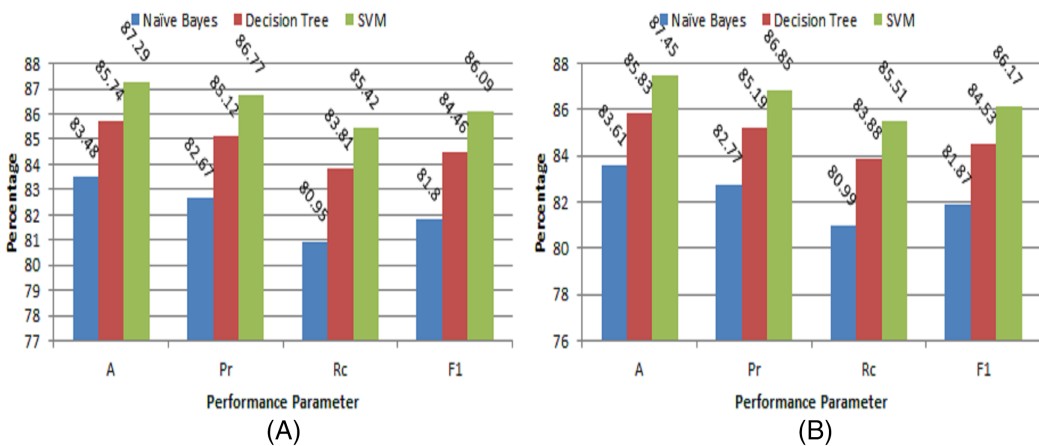

**Figure 5 Performance evaluation of proposed model on our own collected dataset (A) with 70:30 ratio (B) with 5-fold cross-validation.**

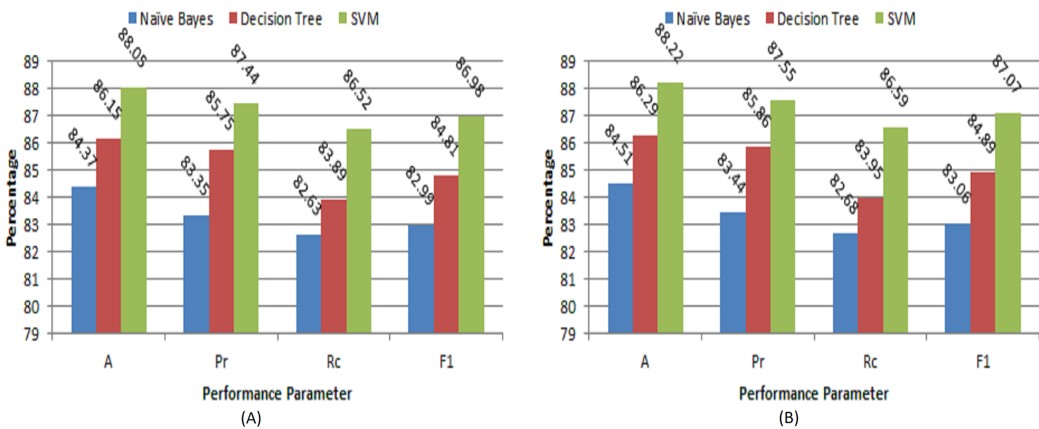

**Figure 6 Performance evaluation of proposed model on NIT Raipur crop dataset (A) with 70:30 ratio (B) with 5-fold cross-validation.**

this automation task. The performance of this system may further improve with experience as new data will be collected. The user feedback will further fine-tune this system even if there are a few wrong suggestions. The experimental results clearly demonstrate that the SVM-based model outperforms other classification models on both datasets. Of the two datasets, the NIT Raipur crop dataset performs better, which may be due to the larger number of samples present in it.

## CONCLUSION AND FUTURE WORK

In this work, a smart irrigation system prototype for efficient usage of water and minimal human intervention is proposed. The proposed recommendation system includes regression of soil and environmental attributes, which are further improved with the help of AC. Forecasted weather data are integrated with these predicted attributes to reduce the irrigation error. Finally, the classification model is used to categorize the combined set of attributes for the suggestion of irrigation is required or not. Based on these results, the

system recommends the farmer for the next irrigation. If the farmer rejects the approval, then feedback is sent to the system. Further, it updates and fine-tunes the model subsequently. The experiment reveals that the SVM model outperforms other classification models on both datasets. As the ML-based models are data hungry, there is a strong intuition that the proposed system will perform even better with more samples. This system can be further extended to deciding on spraying appropriate chemicals for proper growth of crop.

## ACKNOWLEDGEMENTS

The authors wish to thank the Department of Computer Application of NIT Raipur for making available the Crop dataset.

### Funding

This work is financially supported by the Collaborative Research Scheme (CRS) of the National Project Implementation Unit (NPIU), MHRD, Government of India. The funders had no role in study design, data collection and analysis, decision to publish, or preparation of the manuscript.

### Grant Disclosures

The following grant information was disclosed by the authors:
Collaborative Research Scheme (CRS).
National Project Implementation Unit (NPIU), MHRD, Government of India.

### Competing Interests

The authors declare that they have no competing interests.

### Author Contributions

- Ashutosh Bhoi conceived and designed the experiments, performed the experiments, analyzed the data, performed the computation work, prepared figures and/or tables, authored or reviewed drafts of the paper, and approved the final draft.
- Rajendra Prasad Nayak conceived and designed the experiments, performed the experiments, analyzed the data, performed the computation work, prepared figures and/or tables, authored or reviewed drafts of the paper, and approved the final draft.
- Sourav Kumar Bhoi conceived and designed the experiments, performed the experiments, analyzed the data, performed the computation work, prepared figures and/or tables, authored or reviewed drafts of the paper, and approved the final draft.
- Srinivas Sethi conceived and designed the experiments, analyzed the data, performed the computation work, prepared figures and/or tables, authored or reviewed drafts of the paper, and approved the final draft.
- Sanjaya Kumar Panda conceived and designed the experiments, performed the experiments, analyzed the data, performed the computation work, prepared figures and/or tables, authored or reviewed drafts of the paper, and approved the final draft.

- Kshira Sagar Sahoo conceived and designed the experiments, performed the experiments, analyzed the data, performed the computation work, prepared figures and/or tables, authored or reviewed drafts of the paper, and approved the final draft.
- Anand Nayyar conceived and designed the experiments, performed the experiments, analyzed the data, performed the computation work, prepared figures and/or tables, authored or reviewed drafts of the paper, and approved the final draft.

## Data Availability

Data is available at GitHub:

https://github.com/NitrrMCACommunity/Agriculture-Automation-.

https://github.com/GCEKIoTCommunity/Irrigation-Dataset.

## Supplemental Information

Supplemental information for this article can be found online at http://dx.doi.org/10.7717/peerj-cs.578#supplemental-information.

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
