# Peer review of "IoT-IIRS: Internet of Things based intelligent-irrigation recommendation system using machine learning approach for efficient water usage"

_PeerJ Computer Science, doi:10.7717/peerj-cs.578_

## Round 0.1 · original submission · Major Revisions

Please do your best to enhance the paper presentation and respond to the reviewers' comments, all the best.

Reviewer 1 ·

Basic reporting

The literature review in the introduction is very poor.

Experimental design

The authors claim that "a smart irrigation system prototype for efficient usage of water and minimal human intervention has been proposed". However, I am unable to find this prototype in the paper.

Validity of the findings

The authors claim that "a smart irrigation system prototype for efficient usage of water and minimal human intervention has been proposed". However, I am unable to find this prototype in the paper.

Additional comments

This manuscript is about IoT-IIRS: Internet of Things based intelligent irrigation recommendation system using machine learning approach for efficient water usage. As a reviewer, I am unable to accept it due to the following comments:

1. The title is not in accordance with the work done, for example, I am unable to find the implementation of IoT in the manuscript.
2. The literature review in the introduction is very poor.
3. The quality of the figures is very poor.
4. The authors claim that "a smart irrigation system prototype for efficient usage of water and minimal human intervention has been proposed". However, I am unable to find this prototype in the paper.

Based on the above-mentioned reasons, the contribution of this paper is not clear.

Reviewer 2 ·

Basic reporting

The English writing should be improved.
Your introduction needs more detail. I suggest you improve lines 55-57 to show why you think the existing researches are inadequate.
The authors spend a lot of ink on related work (lines 75-129), but how exactly are these works related to the proposed method is not very clear.
Figure 2 should be improved.

Experimental design

The contribution of the paper is too long. I suggest the authors rewrite it to underline the original work they did.
The authors are also suggested to provide more detail on how to implement the proposed system, and analyze its advantages and limitations.

Validity of the findings

The results are not very convincing since the authors fail to analyze why IoT-IIRS is robust, and implement it using experiment.

Additional comments

Dear authors:
After reading your paper, I believe you have good knowledge of intelligent irrigation recommendation system and machine learning. However, the paper needs improvement before accepting. Here are some suggestions:
The writing of the paper is quite fluent except that there are a few grammar mistakes. For instance, the situation may lead to worse in the coming days.
It is difficult to understand the motivation of the paper.
Why some references in the article are in bracket while others are not.

---

## Round 0.2 · Minor Revisions

Dear Dr. Sahoo and Dr. Nayyar,

Thanks for your response. Your manuscript needs to be edited for English before it will be suitable for publication.

All the best

Abdel-Hamid

---

## Round 0.3 · Major Revisions

Thanks for your response, the assigned reviewers have advised clarifying some additional more points, could you please check and respond to their comments.

Reviewer 1 ·

Basic reporting

The paper has been revised, but I didn't understand how authors have implemented IoT using GSM board ?? ! Moreover, there is a lot of "?" in the paper and I didn't understand why?

Experimental design

Why you have used Arduino UNO since there is a lot of embedded board which are didicated ti IoT application such as: Arduino nano IoT 33, ESP32, Raspberry ...

Validity of the findings

The paper has been revised, but I didn't understand how authors have implemented IoT using GSM board ??

Additional comments

The paper has been revised, but I didn't understand how authors have implemented IoT using GSM board ?? ! Moreover, there is a lot of "?" in the paper and I didn't understand why?

Reviewer 3 ·

Basic reporting

The manuscript has some typing error, in Cloud Level sub-section before de equation 1: you have has an extra letter in "literawture" word, please correct it.

Acronyms should be defined where they are first used (e.g. NIT, MSP430, API). In the abstract: NIT is not defined, in the introduction section: MSP 430 is not defined, in the results and discussion section not defined the acronym API, please add it.

Some references are not clear in the manuscript please check it, For instance, (res, 2020) (imd,2020) (rai, 2020), (PME, 2021).

Variables should be defined of the equation 1 (eg. E, Vo, Cf, and Wneed)

Figures 1 to 7 you can improve the legends (number on the bar), changing the size or number style.

In order to clarify, the author should be added a figure of the prototype experimental in real conditions work and figure of the application web.

Experimental design

Crop Field Level subsection, mentions that the microcontroller is connected to the motor. The question is: how is connected to the motor – microcontroller, you use a control circuit or integrated circuit? Can explain this.

How energy consumption of the Crop Field Level sensor, in sending messages, acquisition data, and sleep condition?

Validity of the findings

In my opinion, the strategy implemented in the work is not clear. For instance, Goap et. al. are reported IoT-based smart irrigation using Machine Learning, in my opinion, this work is very similar to your report. Please explain clearly the differences and your contribution.

---

## Round 0.4 · accepted · Accept

Dear Dr Sahoo and Dr Nayyar,

Thanks for making the required amendments. The paper has been accepted now, wishing you all the best.

Reviewer 3 ·

Basic reporting

no comment

Experimental design

no comment

Validity of the findings

no comment

Additional comments

The authors proposed an IoT-enabled machine learning-trained recommendation system for efficient water usage with the nominal intervention of farmers. The authors deployed IoT devices in the crop field to collect the ground and environmental details precisely. The gathered data are forwarded and stored in a cloud-based server, which applies machine learning approaches to analyze data and suggest irrigation to the farmer. The authors added an inbuilt feedback mechanism to this system To make the system robust and adaptive. Finally, the authors performance their IoT devices on the crop, and these results compared with their own collected dataset and NIT Raipur crop dataset is agreed. In my opinion, the authors have attended and corrected the observations and doubts previously issued, so that, the article can be publishable.